# Efficient Subgame Refinement
# for Extensive-form Games

**Zhenxing Ge**
Nanjing University
Nanjing, Jiangsu, China
zhenxingge@smail.nju.edu.cn

**Zheng Xu**
Nanjing University
Nanjing, Jiangsu, China
xuzhengcs@smail.nju.edu.cn

**Tianyu Ding**
Microsoft Corporation
Redmond, Washington, USA
tianyuding@microsoft.com

**Wenbin Li**[*]
Nanjing University
Nanjing, Jiangsu, China
liwenbin@nju.edu.cn

**Yang Gao**[*]
Nanjing University
Nanjing, Jiangsu, China
gaoy@nju.edu.cn

## Abstract

Subgame solving is an essential technique in addressing large imperfect information games, with various approaches developed to enhance the performance of refined strategies in the abstraction of the target subgame. However, directly applying existing subgame solving techniques may be difficult, due to the intricate nature and substantial size of many real-world games. To overcome this issue, recent subgame solving methods allow for subgame solving on limited knowledge order subgames, increasing their applicability in large games; yet this may still face obstacles due to extensive information set sizes. To address this challenge, we propose a *generative subgame solving (GS2) framework*, which utilizes a generation function to identify a subset of the earliest-reached nodes, reducing the size of the subgame. Our method is supported by a theoretical analysis and employs a diversity-based generation function to enhance safety. Experiments conducted on research games as well as the challenging large game of GuanDan demonstrate a significant improvement over the blueprint.

## 1 Introduction

Subgame solving is a standard technique commonly adopted by superhuman AIs to play in games [Campbell et al., 2002, Silver et al., 2016]. This approach takes advantage of the fact that for perfect information games (where both players have complete knowledge about the state of the game), finding optimal strategies at any given decision node only requires knowledge of the subgames rooted at that node [Brown and Sandholm, 2017]. Therefore, AI agents can make use of this fact to improve their own strategies while playing with others, which proves significantly beneficial for large games, *e.g.,* Chess [Campbell et al., 2002] and Go [Silver et al., 2017].

Recent advancements in subgame-solving techniques have proven to be highly successful in imperfect information games, such as no-limit Texas hold'em poker [Moravčík et al., 2017, Brown and Sandholm, 2018, 2019], Mahjoog [Li et al., 2020], and dark chess [Zhang and Sandholm, 2021]. However, state-of-the-art subgame solving algorithms suffer from the efficiency of constructing and solving large games. This limitation hinders their practicality for performing subgame-solving on large subgame trees, thus restricting their application in games such as DouDiZhu [Zha et al., 2021, Jiang et al., 2019] and GuanDan [Lu et al., 2022], which have average information set sizes

---

[*]Corresponding author.

37th Conference on Neural Information Processing Systems (NeurIPS 2023).

up to $10^{23}$ and $10^{30}$, respectively. The substantial size of these games dramatically increases the magnitude of players' $|\mathcal{I}_i|$ in subgame, rendering it impractical to find desirable strategies in real time, as convergence within the limited time frame cannot be guaranteed. To address this computational challenge, techniques such as Abstraction [Sandholm, 2015, Brown et al., 2015] are commonly used in Texas Hold 'em. However, it has been observed that these techniques are inadequate for games like GuanDan [Zha et al., 2021]. A recent work [Zhang and Sandholm, 2021] has partially alleviated this problem by partitioning the subgame tree according to players' knowledge order and discarding nodes that are "far away" from the current infoset. This approach can enable safe strategy refinement under specific conditions of usage in a much smaller subgame, effectively reducing the number of subgame-solving player's information sets $|\mathcal{I}_i|$. Nonetheless, the large number of states within other players' infosets $|\mathcal{I}_{-i}|$ remains unmanageably large and cannot be further reduced, making it difficult to apply these approaches to large games. Moreover, the players' infoset might be complex, and enumerating all states within these sets during subgame construction can be a non-trivial task, adding to the complexity of subgame development.

To address the intricate nature of large games, we present a novel approach called *generative subgame solving (GS2)*. Real-time refinement of these games requires the utilization of a reduced subgame that encompasses fewer players' infosets, which can be efficiently constructed and solved in an online manner while maintaining certain safety guarantees. Our proposed GS2 framework employs a generation function that generates a subset of the other player's infoset, effectively reducing the number of information states $|\mathcal{I}_{-i}|$. Subsequently, a subgame is constructed using nodes rooted in these states. The proposed GS2 framework enables players to construct a more manageable subgame, even in the context of large games. However, the exploitability of the refined strategy may potentially increase compared to the previous knowledge-limited subgame solving, as there is information loss during the construction of the subgame tree. To mitigate this, we begin by providing a theoretical analysis of GS2. We then propose a diversity-based generation function that produces "diverse" subsets, as determined by the analysis, to reduce the exploitability of the refined strategy. Experiments show that GS2, when combined with a diversity-based generation function, significantly enhances performance, making it comparable to the knowledge-limited subgame solving method, while requiring much less computation time.

In this paper, we make the following specific contributions:

- We propose a novel subgame solving method—*generative subgame solving (GS2)*. Unlike traditional techniques that require the full information set, GS2 innovatively generates a subset of the opponent's infoset, thereby reducing the number of information states. This allows for substantial subgame size reduction, making GS2 particularly beneficial for large games, which have historically been challenging due to their vast information set sizes.

- We provide the theoretical analysis of GS2 with arbitrary generation functions, a feature that has been largely overlooked in the past. We leverage the theoretical understanding to introduce a diversity-based generation function for GS2, leading to a significant reduction in the potential exploitability of the refined strategy, a common issue encountered in previous knowledge-limited subgame-solving techniques.

- We conduct extensive experiments for evaluating GS2 on both research games and the notoriously complex large poker game, GuanDan. The experiments underscore the practical value of our method, as GS2 successfully reduces exploitability and increases the expected payoff with appropriate generation functions. In comparison to the knowledge-limited subgame-solving method, GS2 delivers comparable performance while significantly reducing computation time. Moreover, in relatively large games with time limits, GS2 outperforms the knowledge-limited subgame solving, thus showcasing its effectiveness and efficiency.

## 2  Related Work

The concept of refining a coarse strategy (also known as a blueprint) in imperfect information games has been studied for a long time [Gilpin and Sandholm, 2006]. A crucial component of modern refinement techniques is determining the new strategy, which can be adapted from the blueprint or computed from scratch.

The adaptation-based method samples the outcomes of the current games and adapts the current strategy. Online outcome sampling (OOS) method [Lisỳ et al., 2015] is a simulation-based algorithm that relies on Monte Carlo Counterfactual Regret Minimization [Lanctot et al., 2009], which conducts

targeted outcome sampling and updates the regret of each action, resulting in an adapted strategy. Monte Carlo Continual Resolving method [Šustr et al., 2019] combines OOS with the Continual Resolving method [Moravčík et al., 2017] to enhance sampling efficiency while minimizing memory costs associated with OOS. Parametric Monte-Carlo Policy Adaptation method [Li et al., 2020] improves this by performing Monte Carlo tree search for sampling, as well as gradient updates using the sampled trajectories to adjust the blueprint. Despite being theoretically sound, most Adaptation-based Methods are limited in application due to requiring online adaptation to be performed using the same or similar method that computes the blueprint.

On the other hand, methods that compute the new strategy from scratch have no such requirement. Endgame subgame solving methods [Ganzfried and Sandholm, 2015, Gilpin and Sandholm, 2006] first construct the subgame and assume that all the players conform to the pre-defined blueprint before reaching the subgame. The new strategy is then computed in the subgame and the player that performs endgame solving proceeds to implement the new strategy instead of the blueprint in the subgame. Conversely, safe subgame solving methods [Burch et al., 2014, Moravcik et al., 2016, Brown and Sandholm, 2017, Brown et al., 2018, Zhang and Sandholm, 2021] use an evaluation function to construct an augmented subgame, ensuring that the new strategy devised in the augmented subgame is less exploitable than the blueprint.

Search algorithms are also applied for obtaining better joint policy within teammates in collaborative games such as Hanabi [Lerer et al., 2020, Hu et al., 2021] and the bidding phases of contract Bridge [Tian et al., 2020]. For example, SPARTA [Lerer et al., 2020] utilizes exact belief updates for single-agent search and multi-agent search and retrospective belief updates for multi-agent search to handle the large belief range. Subsequently, the strategy is improved through Monte Carlo rollouts. On the other hand, instead of maintaining an explicit belief distribution in single-agent search, Learned Belief Search [Hu et al., 2021] method uses a learned belief model to sample states from the belief distribution, allowing for the application in games with large belief space. Rather than simply improve the strategy via rollouts, Joint Policy Search [Tian et al., 2020] method first decomposes the global changes of the game value into local changes at each infoset, and then iteratively improves the strategy based on the decomposition. Although these approaches primarily focus on improving strategy in collaborative settings and could not be directly used in games with adversaries, the underlying idea could be helpful when developing a new technique that searches within teammates in games like GuanDan.

Our method follows the scheme of subgame solving. Different from the existing methods, our method focuses on expanding the applicability of subgame solving to large games that are intractable for existing techniques. This is achieved by generating a smaller subgame, thereby enabling the integration of techniques that enhance the safety and performance of the refined subgame strategy, such as those presented in [Burch et al., 2014, Brown and Sandholm, 2017].

It is worth noting that previous techniques, such as [Brown and Sandholm, 2015, Brown et al., 2015], also explored the real-time reduction of the entire subgame. These techniques specifically consider the subgame tree that follows a rational opponent's action and disregards the remaining parts. While these techniques appear promising, they lack a solid theoretical guarantee. Furthermore, they are only applicable to *action abstraction*, as the chance player's actions are determined by a fixed probability distribution, and one cannot assume the rationality of the chance player. Thus, these techniques are not suitable for real-world games that involve enormous chance outcomes, such as GuanDan.

## 3 Preliminaries

### 3.1 Extensive-form Games

An extensive-form game with imperfect information can be formulated as $G = \{N, H, P, \mathcal{I}, \{u_i\}\}$, where $N = \{1, 2, 3, \dots\} \cup \{c\}$ is the set of players, $c$ denotes the chance player who chooses actions with a fixed probability, $H$ is the set of all possible history sequences $a^1 a^2 \cdots a^k$ and each $a^i$ is an action. For each history $h \in H$, player function $P : H \to N$ gives the acting player. Additionally, the action set $A(h)$ contains legal actions that player $P(h)$ can choose, which is formally expressed as $A(h) = \{a | h \cdot a \in H\}$. The set of leaf nodes, or terminal nodes, is denoted by $Z$. For each player $i \in N$, the payoff function $u_i : Z \to R$ is the payoff that player $i$ will receive upon reaching a terminal history $z \in Z$. Imperfect information is represented by information sets (infosets).

Infoset $\mathcal{I}$ is the set of information partitions of $H$. For any infoset $I_i \in \mathcal{I}_i$, histories $h, h' \in I_i$ are indistinguishable to player $i$. As such, the acting player and the corresponding legal action set must be the same for all nodes in the same information set. Hence, by denoting the acting player and action set at infoset $I_i$ as $P(I_i)$ and $A(I_i)$, we have $P(I_i) = P(h), A(I_i) = A(h), \forall h \in I_i$. We also use $\mathcal{I}_i(h)$ to denote the infoset that contains $h$ for player $i$.

A strategy $\sigma_i(I_i)$ is a probability distribution over the actions $A(I_i)$ for each infoset $I_i$, and its probability of a specific action $a$ is denoted by $\sigma_i(I_i, a)$. Note that $P(I_i) = i$. Let $-i$ denote all the players in the game except player $i$, $\sigma_i \in \Sigma_i$ and $\sigma_{-i} \in \Sigma_{-i}$ denote player $i$ and $-i$'s strategies, respectively. Player $i$'s expect payoff when all the players adhere to the profile $(\sigma_i, \sigma_{-i})$ is denoted by $u_i(\sigma_i, \sigma_{-i})$. A Nash Equilibrium is a self-enforcing strategy profile $\sigma^*$ that no one has an incentive to unilaterally change its strategy. Therefore, $\forall i \in N, u_i(\sigma_i^*, \sigma_{-i}^*) \geq \max_{\sigma_i' \in \Sigma_i} u(\sigma_i', \sigma_{-i}^*)$. A best response strategy $BR(\sigma_{-i}) = \max_{\sigma_i'} u_i(\sigma_i', \sigma_{-i})$ is a strategy that maximizes player $i$'s own payoff against $\sigma_{-i}$. The exploitability of strategy $\sigma_i$ in a two-player zero-sum game is denoted by $\exp(\sigma_i)$, representing how much payoff player $i$ can lose by changing their strategy from $\sigma_i^*$ to $\sigma_i$ against their opponent's best response strategy. Formally, $\exp(\sigma_i) = u_i(\sigma_i^*, \sigma_{-i}^*) - u_i(\sigma_i, BR(\sigma_i))$, where $\sigma^*$ is the Nash Equilibrium strategy.

For two history $h, h'$, we write $h \sqsubset h'$ if there is an action sequence leading from $h$ to $h'$. The probability of reaching history $h$ when all players act according to $\sigma$ is represented by $\pi^\sigma(h) = \Pi_{h' \cdot a \sqsubseteq h} \sigma_{P(h')}(h', a)$, and $\pi_i^\sigma(h) = \Pi_{h' \cdot a \sqsubseteq, P(h')=i} \sigma_i(h', a)$ is player $i$'s contribution to the probability. The expect value of player $i$ at history $h$ when players play according to $\sigma$ is denoted by $v_i^\sigma(h) = \sum_{z \in Z} \pi^\sigma(h, z) u_i(z)$, where $\pi^\sigma(h, z)$ is the probability of reaching terminal node $z$ from $h$, or 0 if $h \sqsubset z$ does not hold. The value of an infoset $v_i^\sigma(I)$ is the sum of the expected values of all nodes in the infoset, and each one is weighted by the player's counterfactual probability. This can be formally expressed as $v_i^\sigma(I) = \frac{\sum_{h \in I}\left(\pi_{-i}^\sigma(h) v_i^\sigma(h)\right)}{\sum_{h \in I} \pi_{-i}^\sigma(h)}$. The counterfactual best response value $u_i^*(\sigma_{-i}|I)$ to strategy $\sigma_{-i}$ upon reaching infoset $I$ is the best value for player $i$ against $\sigma_{-i}$. This can be defined as $u_i^*(\sigma_{-i}|I) = v_i^{\langle BR(\sigma_{-i}), \sigma_{-i} \rangle}(I)$ and $u_i^*(\sigma_{-i}|Ia) = v_i^{\langle BR(\sigma_{-i}), \sigma_{-i} \rangle}(I, a)$.

An imperfect information subgame (hereafter subgame) is a set of nodes in a game tree, in which information is not divided over infosets. Formally, a subgame $S$ is a set of nodes $h \in H$, such that $\forall h' \in H$, if $\exists h \in S$ and $h' \sqsupset h$, then $h' \in S$. Moreover, $S$ is closed under infoset: $\forall h \in S$, if $h \in I_i$ and $h' \in I_i$ for some player $i$, then $h' \in S$. We define $S_{top}$ as the set of earliest-reachable nodes in $S$, then $h \in S_{top}$ if and only if $h \in S$ and $h' \notin S$ for all $h' \sqsubset h$.

## 3.2 Prior Subgame Solving Methods

In this section, we review the existing methods of subgame solving in imperfect information games. Without loss of generality, we assume that Player 1 (P1) is the responsible party for subgame solving. Within this context, P1 is provided with an initial blueprint strategy and aims to optimize it. During a playthrough, P1 reaches an infoset $I$ and carries out subgame solving to refine the blueprint for the remainder of the game.

As an initial step, standard subgame-solving methods enumerate the top node of the subgame whenever a player reaches a particular information set, allowing them to have direct access to the entire subgame tree. Once the subgame $S$ is constructed, the augmented subgame or gadget game is formed and solved by an appropriate equilibrium finding algorithm. Most subgame solving methods differ mainly in the methodology of constructing the augmented subgame.

The augmented subgame $G'$ in Endgame solving [Ganzfried and Sandholm, 2015] begins with a chance node that leads to all nodes in $S_{top}$. The probability of reaching node $h \in S_{top}$ is proportional to the reach probability of node $h$ in the original game $G$, provided that the players are playing according to the blueprint $\sigma$, that is, reaching $h$ with probability $\frac{\pi^\sigma(h)}{\sum_{h' \in S_{top}} \pi^\sigma(h)}$. The augmented subgame $G'$ is then solved, producing a new strategy $\sigma_1^S$ for P1. For the remainder of the game, P1 is expected to replace the blueprint with $\sigma_1^S$.

Subgame resolving [Burch et al., 2014] differs from endgame solving in the structure of augmented subgames. The initial node of the augmented subgame is a chance node, with Player 2 (P2)'s alternative node, denoted by $h_r$, added between each node $h \in S_{top}$ and the initial node. Each $h_r$ is reached with a probability that is in proportion to P2's probability of reaching $h$, if it tries

to do so–that is $\pi^\sigma_{-2}(h)$. The set of $h_r$ is denoted by $S_r$, with the infoset in $S_r$ being determined according to P2's infoset in $S_{top}$. P2 is presented with two choices for each alternative node: entering $h_{top}$, or terminating the game and receiving an alternative payoff at $\mathcal{I}_2(h_{top})$, which is equal to the counterfactual best response of the blueprint, or $u_2^*(\sigma|\mathcal{I}_2(h_{top}))$.

Subgame resolving develops a strategy that will not be worse than the blueprint. Conversely, Maxmargin subgame solving [Moravcik et al., 2016] seeks to optimize the strategy to ensure that P2 will receive a lower payoff if it chooses to enter the subgame. Formally, the margin of a strategy $\sigma_1^S$ in subgame $S$ is defined as $M(\sigma^S|I_2) = u_2^*(\sigma_1|I_2) - u_2^*(\sigma_1^S|I_2)$. Maxmargin subgame solving hence attempts to find a strategy $\sigma_1^S$ that maximizes $\min_{I_2 \in S_{top}} M(\sigma^S|I_2)$.

On the other hand, knowledge-limited subgame solving (KLSS) [Zhang and Sandholm, 2021] aims to construct a small augmented subgame while providing safety guarantees. KLSS partitions $S_{top}$ into different knowledge orders based on the distance from $I$ in the infoset hypergraph of the subgame. The $k$-th order knowledge limited subgame solving constructs the subgame using nodes at most $k - 1$ units away from the current history $h$ in the original $S_{top}$. KLSS takes advantage of the fact that nodes outside the $k$-th order and their descendants can be discarded if P1's strategy at these nodes is fixed after the payoff is correctly scaled by nodes inside the $(k + 1)$-th order. Hence, the $S_{top}$ of the subgame in the 1st order KLSS (1-KLSS) can be as small as the player's current infoset and its descendants, which significantly reduces the subgame size.

# 4 Generative Subgame Solving (GS2)

## 4.1 General GS2 Framework

In Section 3.2, all methods except KLSS perform subgame solving on the entire subgame tree, which is intractable for large-size games that are difficult to abstract. KLSS attempts to address the problem by limiting the set $S_{top}$ to a lower knowledge order, which reduces the subgame size but is still not feasible for games with a large infoset size. For example, in heads-up Texas Hold'em, where one player only holds two cards, the infoset size is no more than $\binom{52}{2} = 1326$, making it easy to construct for 1-KLSS. On the contrary, in games such as GuanDan, with four players each holding 27 cards at the start, the infoset size can be as large as $10^{30}$. Consequently, constructing and storing the corresponding subgame on a computer becomes impractical.

Even if the subgame is somehow constructed, current equilibrium finding [Lanctot et al., 2009, Farina et al., 2021] algorithm that bound player $i$'s regret by $O(\frac{\sqrt{|\mathcal{I}_i|}}{T})$ is not guaranteed to converge in real-time. This is due to the presence of up to $10^{20}$ other players' infosets in a single player $i$'s infoset in GuanDan. Furthermore, the subgame cannot be abstracted using the techniques in Texas Hold'em [Zha et al., 2021]. Thus, simply applying 1-KLSS would exceed the time limit, necessitating a further reduction in subgames.

Motivated by this, we propose a *Generative Subgame Solving (GS2) method* to reduce the memory and time cost by solving only subgames within a subset of the entire 1-KLSS subgame tree. Specifically, let $\mathcal{Q} = \{Q_1, Q_2, \ldots, Q_R\}$ be the set of all subsets of $S_{top}$, such that if $h \in Q_r, \forall h' \in \mathcal{I}_2(h), h' \in Q_r$. Here, $Q_r$ is referred to as a **block** of $S_{top}$. GS2 then introduces a generation function with probability distribution $f : \mathcal{I}_1 \to \Delta^{|\mathcal{Q}|}$, where $\Delta^{|\mathcal{Q}|}$ is the $|\mathcal{Q}|$-simplex. Upon reaching infoset $I$, GS2 first generates $Q_r \sim f(I)$ and then constructs the generative subgame $S$ such that, $\forall h \in Q_r, h \in S$ and $\forall h' \sqsupset h, h' \in S$.

Let $f(I, Q_r)$ be the probability of $Q_r$ and $\omega(h) = \sum_{r:h \in Q_r} f(\mathcal{I}_1(h), Q_r)$ be the probability that $h$ is considered.

**Proposition 4.1.** *Given a strategy $\sigma_1$ in a two-player zero-sum game $G$, reached infoset $I_1$, and a generation function $f$, the refined subgame strategy $\sigma_1^S$ is computed by GS2 with safe subgame solving techniques. Let $\sigma_1'$ be the strategy that plays according to $\sigma_1^S$ in subgame $S$ and $\sigma_1$ elsewhere. Let $\delta(\sigma_1, h) = \max_{\sigma_1'} u_2^*(\sigma_1'|\mathcal{I}_2(h)) - u_2^*(\sigma_1|\mathcal{I}_2(h))$. Then the exploitability of the expected strategy $\exp(\mathbb{E}[\sigma_1'])$ is bounded by*

$$\exp(\mathbb{E}[\sigma_1']) \le \exp(\sigma_1) + \max_{h \in I_1} (1 - \omega(h)) \, \pi_{-2}^{\sigma_1'}(\mathcal{I}_2(h)) \, \delta(\sigma_1, h). \tag{1}$$

Proposition 4.1 gives the upper bound for GS2 with arbitrary generation functions. $\delta$ can be regarded as a metric for evaluating the effectiveness of the blueprint strategy, measuring its improvement compared to the worst strategy. While GS2 may increase exploitability, it can efficiently reduce the size of the subgame. In balanced games where each unobserved move of other players and chance events have roughly the same size of potential outcomes, only $\frac{|Q_r|}{|S_{top}|}$ of the other player's infoset is considered, thus, the approximate equilibrium can be efficiently computed. Decreasing $|Q_r|$ reduces the computational requirement, but it also leads to lower $\omega(h)$ and a potential increase in exploitability. The choice of the generation function $f$ can be a trade-off between potential improvement and safety.

Intuitively, Proposition 4.1 implies that the new strategies solved by GS2 will be less exploitable if histories $h \in S_{top}$ are more frequently considered in the sampled $Q_r$ (resulting in lower $1 - \omega(h)$) or the blueprint does not perform well (leading to lower $\delta(\sigma_1, h)$). As such, it is suitable to apply GS2 in games with a coarse blueprint, unlike previous unsafe subgame solving methods [Ganzfried and Sandholm, 2015, Gilpin and Sandholm, 2006] that necessitate the blueprint to be close to a Nash equilibrium to construct accurate belief distributions. To further enhance safety, one can apply GS2 in large subgames and switch to other safe subgame-solving methods as the subgame size decreases, since GS2 is intentionally designed to be effective in handling large subgames.

For large-scale games, sampling a block from a non-trivial generation function $f$ is challenging due to the numerous blocks involved. However, randomly generating a single state $h$ in current infoset $I$ is feasible (e.g., randomly choosing a valid opponent hand in poker), and the block can be obtained by repeating the random generation process and adding all $h'$ that is in the same player 2's infoset of $h$. This approach is referred to as the *random generation function*. The generated size $k$ is defined as the size of a randomly generated set of $h$. When using this function, the upper bound of exploitability increment for the expected strategy will decrease in proportion to the size of the generated block. According to Proposition 4.1, we have

$$\exp(\mathbb{E}[\sigma_1']) \leq \exp(\sigma_1) + \left(1 - \frac{k}{|I_1|}\right) \max_{h \in I_1)} \pi_{-2}^{\sigma_1'}(\mathcal{I}_2(h))\delta(\sigma_1, h), \tag{2}$$

for the expected strategy $\mathbb{E}[\sigma_1']$.

To enhance the safety of GS2, it is suggested by Equation (2) to generate as many states as time permits. While increasing the number of states would improve the guarantee, it would also increase the time and resource requirements for solving the subgame. However, the generated size can be controlled and kept relatively small compared to the original subgame. Moreover, techniques such as Abstraction, which traverse the entire infoset, can be combined to further reduce the subgame size while maintaining the theoretical guarantee. It is preferable to have larger generated blocks, which require simpler techniques to operate on them. Previous techniques [Ganzfried and Sandholm, 2014] that involve comparing all potential outcomes would be time-consuming in large games. In contrast, GS2 reduces the subgame size by considering the counterfactual value of the states according to a *diversity-based generation function*. In general, a diversity-based generation function generates a block $Q'$ such that the maximum earth mover's distance between any player 2's infoset and $Q'$ is minimized:

$$Q' = \arg\min_Q \max_{I_2 \subset Q_r \setminus Q} dist(I_2, Q), \tag{3}$$

where $dist(I_2, Q) = \sum_i^N \min_{I_2' \in Q} |u_i^*(\sigma_1|I_2') - u_i^*(\sigma_1|I_2)|$. Let $\mathcal{H}(I_2')$ be the set of infoset that is not vastly dissimilar from $I_2'$, specifically defined as $\{I_2 \in Q_r \setminus Q' | I_2' = \arg\min_{I_2 \in Q_r \setminus Q'} \sum_i^N |u_i^*(\sigma_1|I_2') - u_i^*(\sigma_1|I_2)|\}$. The counterfactual reach probability of infoset $I_2'$ is scaled up by $\sum_{I_2 \in \mathcal{H}(I_2')} \pi_{-2}^{\sigma_1}(I_2)$. In two-player zero-sum games, it is equal to finding a subset such that the alternative values of infosets are likely to be uniformly distributed in range $[\min_{I_2 \subset Q_r} u_2^*(\sigma_1|I_2), \max_{I_2 \subset Q_r} u_2^*(\sigma_1|I_2)]$, which can be efficiently obtained through techniques such as dynamic programming.

Intuitively, the diversity-based generation function generates a block that is the most "diverse" and is able to encompass a wide range of situations (including unobserved others' or chance behaviors) with a small number of infosets. The pseudo-code for GS2 utilizing the diversity-based generation function is presented in Appendix C.8.

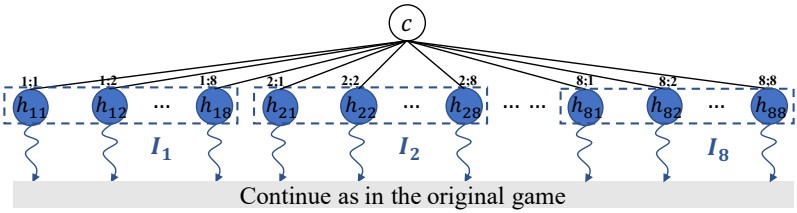

(a) Part of the Liar's Dice game tree.

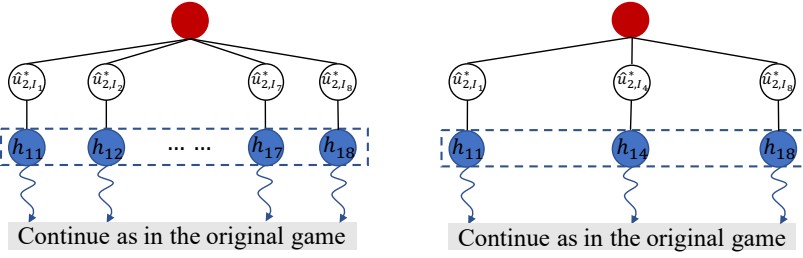

(b) 1-KLSS subgame tree of infoset $I_1$.     (c) GS2 subgame tree of infoset $I_1$.

Figure 1: Example of GS2 in Liar's Dice. The white node is a chance node, representing exogenous events. Nodes are colored in blue if the acting player is player 1 and red for player 2. (a) Part of the Liar's Dice game tree. $h_{i,j}$ is the history that the two players' dice landed on $i$ and $j$, respectively. For simplicity, descendants of $h_{ij}$ that would not be changed in the standard implementation of 1-KLSS and GS2 are omitted. (b) The 1-KLSS subgame tree of infoset $I_1$. (c) The GS2 subgame tree of infoset $I_1$. GS2 only constructs the subgame with a subset of $I_1$. The subset shown is for illustration purposes and would be generated by a generation function in practice.

## 4.2 Example of How GS2 Works

To further clarify the concept of GS2, we present a simple example. Figure 1 illustrates an example of GS2 applied in Liar's Dice with 8-sided dice. In this example, $c$ (the white node) is the chance node, representing exogenous events. Nodes are colored in blue if the acting player is player 1 and red for player 2. Figure 1(a) shows a portion of the Liar's Dice game tree. Each player dice at the beginning. For simplicity, the remainder of the game is omitted and replaced by the gray area, as the nodes rooted at each history $h$ will not be altered in the subgame-solving process.

If player 1 wishes to refine its strategy after determining that infoset $I_1$ has been reached, the subgame constructed by 1-KLSS is shown in Figure 1(b). The chance nodes are redundant since they only have one chance outcome but are retained for consistency. $\hat{u}^*_{2,I_m}$ is the scaled alternative value of $\hat{u}^*_2(\sigma_1|\mathcal{I}_2(h_{1m}))$ for player 1's blueprint $\sigma_1$. Formally, $\hat{u}^*_{2,I_m} = \frac{\hat{u}^*_2(\sigma_1|\mathcal{I}_2(h_{1m})) \sum_{h' \in \mathcal{I}_2(h_{1m})} \pi^{\sigma_1}_{-2}(h')}{\pi^{\sigma_1}_{-2}(h_{1n})}$.
The payoff for player 2 is then subtracted by $\hat{u}^*_{2,I_m}$ upon reaching the corresponding node. Additional payoffs should also be taken into consideration, accounting for the omitted subtree.

GS2 further reduces the subgame size, as shown in Figure 1(c). It only constructs a subgame for a subset of $I_1$, which is generated by a diversity-based generation function. This not only increases the computational efficiency of the subgame solving in large games but also refines the blueprint by selectively focusing on the most typical portions of the game tree.

# 5 Experiments

## 5.1 Evaluation on Research Games

To evaluate the practical performance of GS2, experiments are conducted on a range of commonly used research games with varying infoset sizes.

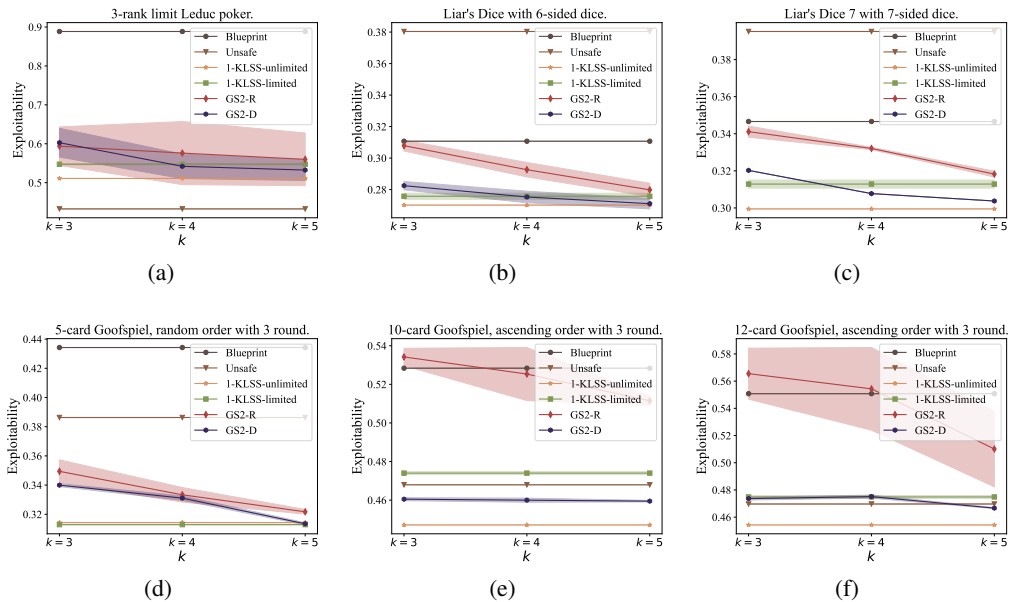

Figure 2: Exploitability of GS2 with random generation function (GS2-R) and diversity generation function (GS2-D) in different games (**Lower is better**). GS2 and 1-KLSS-limited are required to refine the strategy in 3 seconds, while 1-KLSS-unlimited and Unsafe subgame solving method are not. (a) 3-rank limit Leduc poker. (b) Liar's Dice with 6-sided dice. (c) Liar's Dice 7 with 7-sided dice. (d) 5-card Goofspiel, random order with 3 rounds. (e) 10-card Goofspiel, ascending order with 3 rounds. (f) 12-card Goofspiel, ascending order with 3 rounds.

**Implementation details.** All games are implemented using the open-source library Open-Spiel [Lanctot et al., 2019]. In each game, a blueprint is first computed via the CFR algorithm, after which GS2 is applied at each infoset after both players have taken one action to refine the blueprint. The subgame is solved in a nested fashion, utilizing Maxmargin solving with linear programming whenever feasible, and otherwise employing MCCFR [Lanctot et al., 2009]. The unsafe subgame solving method is implemented based on Ganzfried and Sandholm [2015], Brown and Sandholm [2017]. The computation time is limited for 1-KLSS [Zhang and Sandholm, 2021] and GS2. In these methods, the player is allotted a maximum of 3 seconds to compute an approximate Nash Equilibrium after constructing a subgame. In order to provide a comprehensive evaluation of GS2's performance, we also present the results of 1-KLSS and unsafe subgame solving, where there are no time constraints imposed on the computation. It is important to note that GS2 is not expected to surpass the performance of these two methods, as obtaining an exact Nash Equilibrium may require significantly more time. For instance, when solving the subgame of Liar's Dice with 7 dice sides, the computation time for the strategy in unsafe subgame solving can reach up to 122 seconds, and 41 seconds for 1-KLSS, rendering them impractical for real-time decision-making. In contrast, GS2 with $k = 3$ only necessitates 10 seconds of computation time at the same infoset[2], showcasing its efficiency.

**Results.** The effect of varying the value $k$ (*i.e.*, the number of other player's infosets in the generated block) in the diversity-based generation function is evaluated, and the exploitability of blueprints and refined strategies is presented in Figure 2. Experimental results demonstrate that GS2 is capable of effectively reducing exploitability in practice for a coarse blueprint in limited computation time, despite the absence of safety guarantees. As the size of the game increases, GS2 with diversity generation exhibits superior performance compared to 1-KLSS with limited computation time, as evident from the results presented in Figure 2(e) and Figure 2(f). However, in the case of 10-card Goofspiel and 12-card Goofspiel, where the infoset size is extremely large, GS2 with random generation is comparatively inferior to that of other methods. This discrepancy arises due to the inherent limitation of randomly generating a block, as it fails to yield a desirable subgame within the confines of an infoset size reaching up to 1110. Consequently, this inability to generate a good

---

[2]The experiment was conducted on an Intel(R) Xeon(R) Gold 6242R CPU @ 3.10GHz

subgame leads to an increase in exploitability. For 5-card Goofspiel and Liar's Dice, GS2 with diversity generation performs closely to 1-KLSS-unlimited but with significantly less computation time, showing the effectiveness of GS2. Additionally, the results suggest that as $k$ increases, the exploitability of the refined strategy decreases, which aligns with previous theoretical analysis.

For other baselines, the unsafe subgame solving demonstrates excellent performance in 3-rank limit Leduc poker, but it performs inferior in Liar's Dice due to the lack of a safety guarantee. While 1-KLSS without time limits exhibits the best performance in most games, it requires significantly more computation time than GS2, as discussed previously.

## 5.2  Evaluation on GuanDan

We use the techniques in Appendix C.1 to create an agent for the game of GuanDan. The choice to test GS2 on GuanDan is driven by the game's complexity, which surpasses that of other games such as DouDizhu, and the availability of several benchmark agents due to a GuanDan AI competition. GS2 is tested against the two champions of the competition to evaluate its performance:

- The first champion is a rule-based agent and is tested against top human professionals. Despite never having won in a GuanDan tournament, the agent has achieved victory for certain decks in the tournament and has proven to be challenging for experts who are not familiar with it.
- The second champion is a DMC [Zha et al., 2021] agent with warm-starting and post-processing that has a 90% winning percentage against the first champion in the GuanDan tournament. We believe it to be an expert-level agent, although it has not been tested against human experts.

**Implementation details.**   The state-action value function of GuanDan is created by DMC and the subgame is further simplified as outlined in C.2. The approximation is implemented based on the concept of not just sampling from opponent infosets but also within the infoset itself to manage complexity. This approach is like applying a specific generation function to the opponent's infoset. It leads to a conservative strategy refinement by assuming opponents are aware of the player's exact private information. Such an assumption prevents a drastic rise in exploitability at the present infoset. This method is influenced by patterns observed in the game of GuanDan. In this game, as actions unfold, uncertainty diminishes, thus human players are expected to guess the others' cards, especially near the end of the game, which is crucial for expert players. Therefore, considering the worst case that the opponents know the player's cards would be practical. Even though this idea seems somewhat domain-specific, we believe it is also applicable to similar games.

During the entire game, only one player performs subgame solving, while the teammate continues to play according to the state-action value function. In other words, the teammate will select the action with the highest value among all legal actions. The opponent team consists of two identical agents who do not communicate during the game. To better evaluate performance, we conduct tournaments with a fixed game level. In each tournament, a deck is played four times to minimize variance. The player who performs GS2 rotates their seat to the next player after each round of play. At the end of each game, players will receive a score according to the order in which the player and its teammate empty their hand, which is equal to the value of levels they would prompt in the original GuanDan game for the winning team and the negative value for the losing team. A more detailed description of GuanDan is given in Appendix B. The tournament is then repeated 100 times and the average score of GS2 in each position is calculated.

**Results.**   We first present the results when playing against the second champion. Different settings of $k$ are applied in the tournament. Despite the remarkably small value of $k$ relative to the size of the infoset, the results shown in Figure 3(a) indicate that the application of GS2 leads to a significant performance gain. It also shows that as more states are generated by GS2, the better its performance becomes. The results of GS2 against the first champion are given in Figure 3(b). Compared to the first champion, GS2 has less improvement in the average score. We believe that this is attributed to the outstanding performance of the strategy constructed by the value function, which results in the improvement not being fully reflected in the average score. Overall, the experiment results suggest that GS2 is able to improve the performance of a given blueprint in large games with desirable subgame sizes.

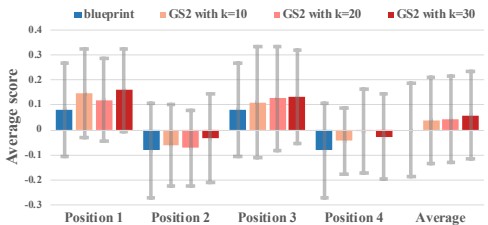
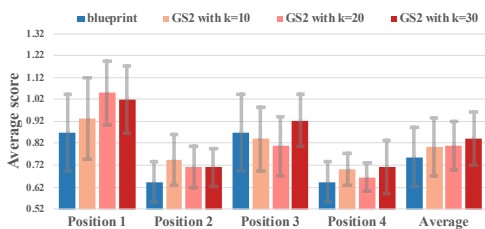

| (a) Performance against the second champion agent. | (b) Performance against the first champion agent. |

Figure 3: The avarage score of GS2 against two prior agents at each position in GuanDan (**Higher is better**). The score of the blueprint (blue) gives the baseline value that agents could achieve without subgame solving at each position.

## 6 Conclusion and Future Work

We propose a novel subgame-solving method, named GS2, which is designed to address the challenges posed by games with imperfect information and large infoset sizes. GS2 only constructs a partial subgame tree, thereby reducing the memory and time costs associated with conventional subgame solving. Our method is able to construct robust strategies for large games with unmanageable infoset sizes. We theoretically analyze the exploitability bound for GS2 and propose a diversity-based generation function to mitigate the issue of unsafeness. Through experimental evaluations in research games and a complex poker game–GuanDan, we demonstrate that GS2 indeed decreases exploitability and improves the performance of the given blueprint in practice.

While GS2 shows promising performance with a simple diversity-based generation function, it is important to note that further research is needed to address the limitations of the method, particularly when alternative values are not adequate for describing the state. Additionally, it remains an open question as to whether a safe generation function exists that can further expand the applicability of GS2.

## 7 Acknowledgement

We thank the anonymous reviewers for their constructive comments. This work was supported in part by Science and Technology Innovation 2030 New Generation Artificial Intelligence Major Project(No.2018AAA0100905), the National Natural Science Foundation of China(No. 62192783, No.62106100, No.62276142), Primary Research & Development Plan of Jiangsu Province (No.BE2021028), Jiangsu Natural Science Foundation (BK20221441), Jiangsu Provincial Double-Innovation Doctor Program (JSSCBS20210021), and in part by the Collaborative Innovation Center of Novel Software Technology and Industrialization.

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
