# A Proofs

## A.1 Proof of Proposition 4.1

*Proof.* Let $\langle Q_1, Q_2, \ldots, Q_R \rangle$, $\langle S^1, S^2, \ldots, S^R \rangle$ be all potential blocks and corresponding subgames. Let $\langle \sigma_1^{S^1}, \sigma_1^{S^2}, \sigma_1^{S^3}, \ldots, \sigma_1^{S^k} \rangle$ be all refined subgame strategies. The expected refined subgame strategy $\mathbb{E}[\sigma_1^S]$ is the average strategy of $\langle \sigma_1^{S^1}, \sigma_1^{S^2}, \sigma_1^{S^3}, \ldots, \sigma_1^{S^k} \rangle$ weighted by $f$, denoted by $\bar{\sigma}_1^S$. Hence, for each history $h \in S_{top}$, the P2's counterfactual best response value

$$u_2^*(\bar{\sigma}_1^S | \mathcal{I}_2(h)) = v_2^{\langle \bar{\sigma}_1^S, BR(\bar{\sigma}_1^S) \rangle}(\mathcal{I}_2(h))$$
$$= \sum_{i=1}^{R} f(I, Q_i) v_2^{\langle \sigma_1^{S^i}, BR(\bar{\sigma}_1^S) \rangle}(\mathcal{I}_2(h)).$$

Obviously,

$$v_2^{\langle \sigma_1^{S^i}, BR(\bar{\sigma}_1^S) \rangle}(\mathcal{I}_2(h)) \leq v_2^{\langle \sigma_1^{S^i}, BR(\sigma_1^{S^i}) \rangle}(\mathcal{I}_2(h))$$
$$= u_2^*(\sigma_1^{S^i} | \mathcal{I}_2(h)).$$

Then, we have

$$u_2^*(\bar{\sigma}_1^S | \mathcal{I}_2(h)) \leq \sum_{i=1}^{R} f(I, Q_i) v_2^{\langle \sigma_1^{S^i}, BR(\sigma_1^{S^i}) \rangle}(\mathcal{I}_2(h))$$
$$= \sum_{i, h \in Q_i} f(I, Q_i) v_2^{\langle \sigma_1^{S^i}, BR(\sigma_1^{S^i}) \rangle}(\mathcal{I}_2(h)) + \sum_{i, h \notin Q_i} f(I, Q_i) v_2^{\langle \sigma_1^{S^i}, BR(\sigma_1^{S^i}) \rangle}(\mathcal{I}_2(h)).$$

Note that the subgame margin of two strategy $\sigma_1, \sigma_1^{S^i}$ in $S^i$ at $I_2$ is $M(\sigma_1, \sigma_1^{S^i}, I_2) = u_2^*(\sigma_1 | I_2) - u_2^*(\sigma_1^{S^i} | I_2) \geq 0$. Thus, $u_2^*(\sigma_1^{S^i} | I_2) \leq u_2^*(\sigma_1 | I_2)$ and

$$u_2^*(\bar{\sigma}_1^S | \mathcal{I}_2(h)) \leq \sum_{i, h \in Q_i} f(I, Q_i) u_2^*(\sigma_1 | \mathcal{I}_2(h)) + \sum_{i, h \notin Q_i} f(I, Q_i) u_2^*(\sigma_1^{S^i} | \mathcal{I}_2(h)).$$

Recall that

$$\omega(h) = \sum_{r : h \in Q_r} f(\mathcal{I}_1(h), Q_r).$$

We have $\sum_{i, h \in Q_i} f(I, Q_i) = \omega(h)$ and $\sum_{i, h \notin Q_i} f(I, Q_i) = 1 - \omega(h)$.

Then,

$$u_2^*(\bar{\sigma}_1^S | \mathcal{I}_2(h)) \leq \omega(h) u_2^*(\sigma_1 | \mathcal{I}_2(h)) + (1 - \omega(h)) \max_{\sigma_1'} u_2^*(\sigma_1' | \mathcal{I}_2(h))$$
$$\leq u_2^*(\sigma_1 | \mathcal{I}_2(h)) + (1 - \omega(h))(\max_{\sigma_1'} u_2^*(\sigma_1' | \mathcal{I}_2(h)) - u_2^*(\sigma_1 | \mathcal{I}_2(h))).$$

Let $\delta(\sigma_1, h) = \max_{\sigma_1'} u_2^*(\sigma_1' | \mathcal{I}_2(h)) - u_2^*(\sigma_1 | \mathcal{I}_2(h))$,

$$\exp(\bar{\sigma}_1^S | \mathcal{I}_2(h)) \leq \exp(\sigma_1 | \mathcal{I}_2(h)) + (1 - \omega(h)) \delta(\sigma_1, h).$$

Hence, for $h \in S_{top}$ such that $\pi_2^{BR(\mathbb{E}[\sigma_1'])}(\mathcal{I}_2(h)) > 0$,

$$\exp(\mathbb{E}[\sigma_1']) \leq \exp(\sigma_1) + \max_h \pi_{-2}^{\sigma_1'}(\mathcal{I}_2(h))(1 - \omega(h)) \delta(\sigma_1, h).$$

$\square$

# B  Introduction of GuanDan

GuanDan is a popular card game in China. It requires players to cooperate and compete with each other in an imperfect information setting. GuanDan is played among 4 players. The player who sits on the opposite side is the teammates and those who sit downside and upside are opponents. The goal of each player is to put in all their cards to beat the opponents each round and increase the team's level to A to win the whole game.

GuanDan uses two decks of cards, including the four jokers. In other words, each player has 27 cards at the beginning of each game. The brief rules are described as follows.

- **Level:** Each GuanDan game can be divided into several rounds. The current level in each round is determined by the highest level of the two teams. The level of both sides will be updated according to the outcome of each round. The initial level of the two teams is 2. There are 13 levels in total, matching the card ranks from 2 to A.

- **Leveling Up:** After the end of each round, only the team of the player who has finished playing cards first (Banker) can level up. The team could level up by 3 levels if the other teammate finished second (Follower), 2 levels if the other teammate finished third (Third), and 1 level if the other teammate finished last (Dweller).

- **Playing:** Players play cards in counterclockwise order. Each player can only choose to throw out cards the same type as the cards thrown by the previous player or throw a bomb. If no cards are thrown, the player must say "Pass". If the other three players pass for a particular card thrown by the player, then this player can throw any type of card he wants.

- **Winning Condition:** A team can win the whole game if it is in A level this round and then win this round without Dweller.

- **Level Card:** Cards of the same level as the current level are level cards, among which the Heart cards are called wild cards. Level cards are smaller than jokers but larger than any other cards in the corresponding round. Wild cards can be regarded as any cards combined with other cards.

- **Card Type:** In GuanDan, the types of cards are as follows: Single(single card), Pair(two single cards with the same number), Triple(three single cards with the same number), Plate(two consecutive triples), Tube(three consecutive triples), Full House(a triple with a pair), Straight(five consecutive single cards), Bomb(at least four cards with the same number), Straight Flush(a Straight with the same suit in every single card), Joker Bomb(four jokers).

- **Card Score:** In GuanDan, the cards are ranked by Joker, A, K, Q, J, 10, 9, 8, 7, 6, 5, 4, 3, 2. This comparison works for Single, Pair, Triple, Tube, Plate, Straight, Bomb, and Straight Flush. When A is matched into Tube, Plates, Straight, and Straight Flush, it can be regarded as 1. For the Bomb, the greater number of cards means the greater card scores. A straight flush is greater than any bomb with fewer than 6 cards.

- **Tribute:** Starting from the second round, the Dweller of the previous round pays tribute to the Banker, giving the largest card except for the Heart card to the Banker. As a return, Banker chooses any one card not greater than 10 to pay back to the Dweller. If two players in one team are Third and Dweller in the previous round(Double-Dweller), both of them pay tribute to the opposing team and accept the back tribute.

# C  Further Implementation Details and Pseudo-code

## C.1  Implementaion Details of Experiments in GuanDan

In this section, we describe how we use GS2 to refine the strategy in GuanDan, a variant of DouDizhu. The game is characterized by its four players, divided into two teams, and its use of two packs of cards. The significant size of the game makes it challenging to compute a low-exploitable strategy. State-of-the-art techniques have thus far been limited to constructing an infoset-action value function, denoted as $Q(I, a)$, rather than a blueprint $\sigma(I, a)$.

## C.2 Implementation of Diversity-based Generation Function in GuanDan

In GuanDan, the infoset size can reach up to $10^{30}$, rendering it infeasible to even traverse one infoset. To mitigate this issue, we employ diversity generation when generating the subgame, in order to ensure that the subgame can be constructed and solved in real-time. Specifically, when an infoset $I$ is reached, GS2 first randomly samples $m = 1000$ histories $h$, and then calculates a "diverse" subset of $k$ elements from the sampled histories. Although standard GS2 needs to generate the full second-order knowledge infosets, we only consider the nodes in the first-order knowledge infoset. The construction of second-order knowledge infosets is difficult. It takes $2k\times$ time to generate a second-order knowledge infoset for each node which may exceed the time limits.

## C.3 Construction of subgame in GuanDan

Constructing a subtree rooted at a single node in the GuanDan game can prove to be a challenging task, given the depth of the game tree, which can reach up to 400, and the vast action space of approximately $10^6$. In order to alleviate this issue, a key technique known as depth and branch-limited construction is applied. While this technique is described in a game-specific context, it is adaptable to other games as well.

Standard depth-limited subgame solving constructs a trunk that is a modified version of the original subgame, in which certain internal nodes are replaced by terminal nodes [Brown et al., 2018, Zhang and Sandholm, 2021]. At each terminal node, all players have the option to select from a variety of continuation strategies for the remainder of the subgame, known as multi-value states. However, due to a lack of techniques for constructing diverse blueprints for GuanDan, the calculation of these continuation strategies is a complex task, and thus, we resort to using a single approximate infoset-action value function, $Q_i(I, a)$. Formally, the approximate payoff value, $\tilde{u}_i(h)$, of a leaf node, $h$, is obtained via Monte Carlo roll-outs where each player plays according to a strategy that always chooses the action with the highest $Q_i(I, a)$. The default depth limit is set at 10, inclusive of the teammate's node which possesses only a single legal action. However, if the combined card count for all opponents' hands exceeds 27, the depth limit is reduced to 8. In addition, the number of actions in the subgame is limited, with only the $n = 2$ highest-value actions in $Q(I, a)$ for each action type at infoset $I$ considered valid.

## C.4 Single-Player Solving in GuanDan

In order to minimize the computation complexity and ensure prompt decision-making, single-player solving is employed, despite the potential performance improvement that may result from subgame solving for both players in the team. Given that, only one player is performing subgame solving, nodes in which the teammate acts can be replaced by a chance node that selects actions based on the blueprint. This technique allows for a reduction in computation time, while still retaining a significant level of performance.

## C.5 Structure of the Constructed Subgame in GuanDan

The construction of the gadget game used in maxmargin subgame solving Moravcik et al. [2016] is non-trivial in GuanDan due to the increased number of players. Although the opponents share the same payoff for a specific history, their counterfactual best response value $u_i^*$ could be different, making the gadget game's construction infeasible. Thus, GS2 opts to construct a resolving subgame when two opponents are playing and a maxmargin gadget game when only one opponent is playing.

For the resolving subgame, alternative nodes for both opponents are added between the chance node and $S'_{top}$ in the augmented game. Specifically, the chance player moves first, leading to an alternative node for one opponent with options to either *enter* the subgame with the chance outcome or *opt out*. If the opponent chooses to enter, the other opponent will also have the opportunity to choose.

For the maxmargin gadget game, the construction is similar to previous works Moravcik et al. [2016] since only two players are allowed to search in this subgame.

A simple blueprint, wherein players will act according to $a = \arg\max_{a \in A(I)} Q(I, a)$, is used to compute the counterfactual probability, which is proportionally assigned to the chance outcomes.

### C.6 Dealing with Invalid Particles in GuanDan

In the GuanDan experiments, GS2 generates a set $S_r$ from $S_{top}$ with $|S_r| = 1000$ and computes $S'_{top}$ such that $|S'_{top}| = k$ in the sampled subset. However, upon reaching a new infoset $I$, it is likely that some of the histories in $S'_{top}$ are invalid due to the new observations of other players' actions in GuanDan. When $|S'_{top}| < k$, GS2 adds new histories to $S_r$ in order to maintain $|S'_{top}| = k$ if time permits.

According to the diversity-based generation function, GS2 must regenerate $m$ histories and then obtain the corresponding subset. However, this would lead to a complete reconstruction of the subgame tree, which contradicts the intent of nested subgame solving. In practice, GS2 retains the valid $S'_{top}$ and identifies a new history $h'$ in $S_r$ that maximizes the difference between the history and the valid $S'_{top}$. Consequently, $h'$ is thereby added into $S'_{top}$, and the procedure is repeated until $|S'_{top}| = k$ or the time limit is reached. Intuitively, the new $S'_{top}$ is well distributed while it may not be the most diverse subset of the new $S_r$.

Despite the invalid histories from the observation, histories may also be unsupported by the players' strategies. This issue is exacerbated as more actions are taken. Ideally, we might desire $S_r$ to consist of histories with positive counterfactual probability by repeating generating histories and eliminating histories that are not supported. However, repeated generation would be time-consuming and undermine the applicability of the method. Fortunately, GS2 does not assume the counterfactual probability of the history and the histories with zero probability do not necessarily need to be eliminated. In spite of the presence of zero-probability histories, only positive-probability histories are considered for the diversity-based functions.

### C.7 Estimation of the Counterfactual Best Response Value

When constructing the subgame, standard subgame solving methods compute the counterfactual best response value in the constructed subgame Brown and Sandholm [2017], Burch et al. [2014]. However, the incompleteness of the opponent's infoset would lead to incorrectness for the opponent's counterfactual value. Due to the lack of techniques to calculate the exact counterfactual best response value, the value function $Q(I, a)$ is used instead. Specifically, the estimation $\tilde{u}_i^*(\sigma_1|I) = \max_a Q(I, a)$ is used for the reached infoset $I = \mathcal{I}_i(h)$.

### C.8 Algorithms

We give the pseudo-code of the implemented GS2 and diversity-based generative function tailored for research games, as outlined in Algorithm 1 and Algorithm 2.

Algorithm 2 exploits the fact that the first $k$ elements in the sorted diverse set $\mathcal{D}''$, derived from the original set $\mathcal{D}'$, constitute a diverse set $\mathcal{D}''_k$ for the subset $\mathcal{D}'_k$ of $\mathcal{D}'$, where $\mathcal{D}'_k = \{(i, u_2^*(\sigma_1|\mathcal{I}_2(h_i)))|\forall (i, u_2^*(\sigma_1|\mathcal{I}_2(h_i))) \in \mathcal{D}', \min_{v \in \mathcal{D}'_k} v[1] \leq u_2^*(\sigma_1|\mathcal{I}_2(h_i))) \leq \max_{v \in \mathcal{D}'_k} v[1]$.

## D Discussion of the single-player search

As discussed in Appendix C.1, GS2 applies single-player search in GuanDan to ensure prompt decision-making. While allowing both players to search in the subgame for an $\epsilon$-TME or $\epsilon$-TMECor may potentially enhance the final payoff, it would dramatically increase the complexity of subgame construction and resolution.

In the case of TMECor, the team common knowledge subgame must be considered since the teammate lacks knowledge of our agent's private information. This necessitates GS2 to generate histories that are valid for public information, regardless of the private information held by both players. In GuanDan, this equates to generating histories with the observed action history. Additional coordination prior to the game, such as exchanging the global seed, is required to aid the player in predicting the teammate's generated strategy. However, the solved strategy might be problematic because it may not contain either player's infosets, which implies that the constructed subgame refines the strategy at some other infosets and disregards the current one.

---

**Algorithm 1** Generative subgame solving with diversity-based generation functions

---

**Input:** game $G$, blueprint $\sigma_1$, and reached infoset $I$

▷*First construct the Maxmargin subgame.*

Initialize $S \longleftarrow$ empty game

Compute the counterfactual best response value $u_2^*(\sigma_1|\mathcal{I}_2(h)$ for each history $h \in I$

Create root node $\emptyset$ in $S$, where player 2 acts

Call diversity-based generation function to get $S'_{top}$

**for** each player 2's infoset $I_0$ with $I_0 \cap S'_{top} \neq \emptyset$ **do**

    $D = \sum_{h \in I_0 \cap S'_{top}} \pi_{-2}^{\sigma_1}(h)$

    Create chance node $\emptyset I_0$ in $S$

    ▷*Build the subtree rooted at $I_0$. The subtree is different with 1-KLSS. GS2 add all the node in $I_0 \backslash I$ and their descendants into $S$ to enable the application of algorithm such as CFR [Zinkevich et al., 2007].*

    **for** each $h \in I_0 \cap S'_{top}$ **do**

        Copy $h$ into $S$ as a child of $\emptyset I_0$ with probability $\pi_{-2}^{\sigma_1}(h)/D$

    **end for**

    **for** each $h \in I_0 \backslash S'_{top}$ **do**

        Copy $h$ into $S$ as a child of $\emptyset I_0$ with probability $\pi_{-2}^{\sigma_1}(h)/D$

        **for** each $h' \sqsupseteq h$ with $P(h') = 1$ **do**

            ▷*1-KLSS assumes the player's strategy at nodes outside the 1st order knowledge is fixed, which is equivalent to a chance node.*

            Change $h$ to a chance node and the probability of chance outcome $a \in A(h)$ is $\sigma_1(h, a)$

        **end for**

    **end for**

    **for** each terminal node $z \in S$ **do**

        ▷*Subtract the alternative value.*

        $u_2(z) = u_2(z) - u_2^*(\sigma_1|\mathcal{I}_2(z)$

    **end for**

**end for**

▷*Solving the subgame.*

Refined strategy $\sigma_1^S \longleftarrow$ apply subgame solving at $S$

**Return** $\sigma_1^S$

---

In the case of TME, the situation is more complex since the player would be unaware of the teammate's generation, necessitating the consideration of a full 1-team-knowledge-limited subgame. This approach is infeasible for large games.

# E Further Experiments and Details

## E.1 Research Games

For research games, the GS2 with diversity generation function (GS2-D) is implemented by applying Equation (3) with $Q_r = S_{top}$. Thus the GS2 with random generation function (GS2-R) serves as the ablation study of GS2-D to show the effectiveness of the diversity generation function.

We also conduct experiments with different time limits for 1-KLSS and GS2, as outlined in Figure 4 and Figure 5.

It shows that when there is enough time to construct and compute the equilibrium, GS2-D may not expected to outperform 1-KLSS in research games.

## E.2 GuanDan

To illustrate the effectiveness of the diversity-based generation function in GuanDan, we conduct further ablation studies. As depicted in the results presented in Table 1, GS2-R is not able to outperform the blueprint. This outcome can be attributed to the substantial number of states within

**Algorithm 2** Dynamic programming for diversity-based generation functions.

---

**Input:** set of the history index and value tuple $\mathcal{D} = \{(i, u_2^*(\sigma_1 | \mathcal{I}_2(h_i)))\}$, number $n$ of genereted states.

$\mathcal{D}' \leftarrow$ sorted according to the value of $\mathcal{D}$

$m \leftarrow$ size of $\mathcal{D}'$

▷*result$[i][j]$ stores the partition of $\mathcal{D}'[0 : i]$ with $j$ diverse elements with cost$[i][j]$ stores the diversity measure (lower is better).*

Initialize array $result[m, n] \leftarrow -1$, $cost[m, n] \leftarrow \infty$

**for** $i = 0$ to $m - 1$ **do**
  $result[i][1] = 0$
  $cost[i][0] = 0$
**end for**

▷*The transition function is* $cost[i][j] \leftarrow \min_{k \in j-1, j, \ldots, i-1} cost[k][j - 1] + dist(\mathcal{D}'[i][1], \mathcal{D}'[last][1])$

**for** $i = 1$ to $m - 1$ **do**
  **for** $j = 1$ to $\min(i + 1, n) - 1$ **do**
    **for** $k = j - 1$ to $i - 1$ **do**
      **if** $j == 1$ **then**
        $last = 0$
      **else**
        $last = k$
      **end if**
      $total\_cost \leftarrow cost[k][j - 1] + (\mathcal{D}'[last][1] - \mathcal{D}'[i][1])^2$
      **if** $total\_cost < cost[i][j]$ **then**
        $cost[i][j] = total\_cost$
        $result[i][j] = last$
      **end if**
    **end for**
  **end for**
**end for**

▷*Restore the diverse set from result*

$previous = m - 1$

diversity set $\mathcal{D}'' \leftarrow \emptyset$

**for** $i = 0$ to $n - 1$ **do**
  add $\mathcal{D}'[previous]$ into $\mathcal{D}''$
  $previous \leftarrow result[previous][n - i - 1]$
**end for**

**Return** $\mathcal{D}''$

---

Table 1: Performance of GS2-R with $k = 10$ against two prior agents at each position in GuanDan.

| | Average Score Against Second Champion | Against First Champion |
|---|---|---|
| Blueprint | $0.000 \pm 0.156$ | $0.809 \pm 0.128$ |
| GS2-R | $-0.014 \pm 0.182$ | $0.770 \pm 0.121$ |

the current infoset. The simplistic approach of randomly selecting states may lead to an overly optimistic or pessimistic assessment, consequently resulting in an inferior strategy.

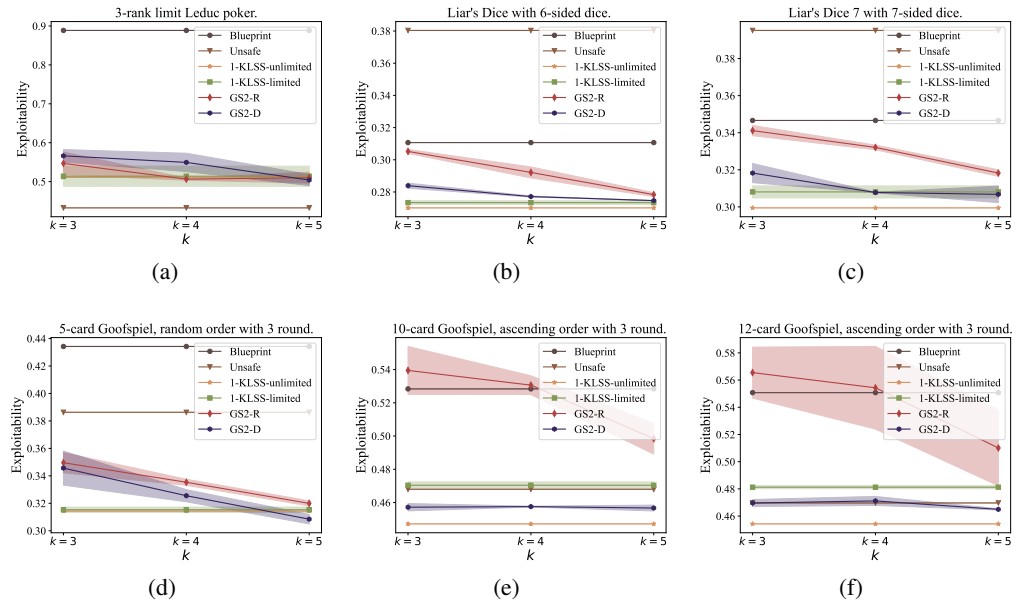

Figure 4: Exploitability of GS2 with random generation function (GS2-R) and diversity generation function (GS2-D) in different games (**Lower is better**). GS2 and 1-KLSS-limited are required to refine the strategy in 4 seconds, while 1-KLSS-unlimited and Unsafe subgame solving method are not. (a) 3-rank limit Leduc poker. (b) Liar's Dice with 6-sided dice. (c) Liar's Dice 7 with 7-sided dice. (d) 5-card Goofspiel, random order with 3 rounds. (e) 10-card Goofspiel, ascending order with 3 rounds. (f) 12-card Goofspiel, ascending order with 3 rounds.

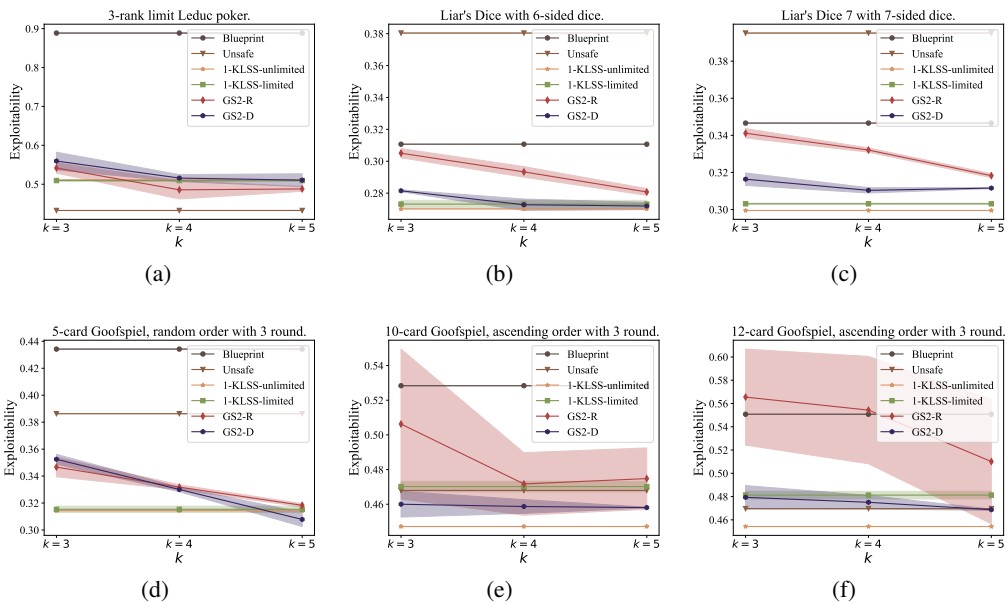

Figure 5: Exploitability of GS2 with random generation function (GS2-R) and diversity generation function (GS2-D) in different games (**Lower is better**). GS2 and 1-KLSS-limited are required to refine the strategy in 5 seconds, while 1-KLSS-unlimited and Unsafe subgame solving method are not. (a) 3-rank limit Leduc poker. (b) Liar's Dice with 6-sided dice. (c) Liar's Dice 7 with 7-sided dice. (d) 5-card Goofspiel, random order with 3 rounds. (e) 10-card Goofspiel, ascending order with 3 rounds. (f) 12-card Goofspiel, ascending order with 3 rounds.