# OpenReview forum: "Efficient Subgame Refinement for Extensive-form Games"
_NeurIPS.cc/2023/Conference — NeurIPS 2023 poster_

### Official Review · Reviewer_gWJJ · 2023-06-20

**Soundness:** 3 good
**Presentation:** 3 good
**Contribution:** 3 good
**Rating:** 6
**Confidence:** 4

**Summary:**

The paper proposes GS2, a method to overcome the problem of large information states in subgame solving in imperfect-information games. Theoretical results and experimental evaluation on GuanDan are presented, with impressive results.

**Strengths:**

The main idea of the paper, to use a diversity function to filter the set of states, is nice. The practical results are particularly impressive: experiments on medium-sized games seem to clearly indicate that the diversity function has a positive impact, and the experiments on GuanDan seem to suggest that the bot is state-of-the-art.

**Weaknesses:**

I was a reviewer on an earlier version of this paper, during which time I had a robust discussion with the paper authors. The present submission is much improved and has addressed most of the concerns I had with the previous version. I have only a few lingering questions and minor concerns from that discussion, mostly concerning the experiments; see below.

**Questions:**

1. (L522) "Although standard GS2 need to generate the full second-order knowledge infosets, we only consider the nodes in the first-order knowledge infoset" Does this imply that, in the subgame, the opponent essentially has complete information about the history at the root of the subgame?
    * If so, this seems like a fairly strong simplification, and I am fairly surprised at the strong practical performance. Why do you believe that GuanDan has such a structure that this simplification does not completely destroy performance?
    * If not, I assume you will have to incorporate second-order nodes *somewhere*. How, precisely is that done?
1. (Sec C.3) What depth and branching factor limits are used in the GuanDan experiments?
1. In the tabular experiments, it appears that the performance of GS2-R and GS2-D get closer to each other as $k$ increases. I would be interested to see experiments with slightly larger $k$, perhaps $k=5$ or $k=10$, to see if this effect continues. Along these lines, how computationally expensive is it in GuanDan to use a larger $k$? That is, if you were to set $k=10$ (say) in GuanDan, would the time taken in subgame solving be then too long to use in realtime?

**Limitations:**

Yes

---

> ### Author Rebuttal · Authors · 2023-08-09
>
> Thanks for your appreciation and valuable comments/suggestions for our works. We hope these response would address the lingering concerns.
>
> * **Regarding  Question 1**.
>
>   Thank you for your insights. In GuanDan, we indeed operate under the assumption that opponents possess complete information. This stems from a sampling across the opponent's infoset, functioning as a specialized sampling approach. The rationale is based on the observation that, as GuanDan progresses, actions reduce game uncertainty. Hence, adept players often anticipate other players' cards -- a crucial skill for proficient players, especially as the endgame approaches.
>
>   It's worth noting that while this simplification might yield a more conservative strategy at the beginning, the subgame is solved in a depth-limited manner. Consequently, leaf node values are derived from a blueprint devoid of complete information. This ensures that opponents cannot excessively capitalize on having complete information, thus mitigating the potential negative impact of our simplification.
>
> * **Regarding Question 2**.
>
>   Thank you for the question.
>
>   \- Branch Factor: We set the branch factor to 2 for each action type. Actions in GuanDan are divided into several types, as detailed in the "Card Type" section of Appendix B. For each infoset, the player is allowed to choose from all types in its hand if no one plays any cards after the player's last action. Otherwise, only the same card types, "Pass (not playing any cards)" or "Bomb (a special cards type)" can be played. Thus the numbers of actions after pruning may not be the same for each infoset.
>
>   \- Depth Limit: The default depth limit is set at 10, inclusive of the teammate's node which possesses only a single legal action. However, if the combined card count for all opponents' hands exceeds 27, the depth limit is reduced to 8.
>
> * **Regarding Question 3**.
>
>   Thank you for your insightful question.
>
>   As $k$ grows larger, the sets generated by GS2-R are more likely to be diverse in small infoset, thus yielding similar performance outcomes when contrasted with those generated by GS2-D. As you suggested, we conduct experiments with $k=8$ and $k=10$ in Goofspiel. The results for these additional experiments can be found in Section 2 of the global response PDF. The results shows the performance of GS2-D and GS2-R would getting closer as $k$ increasing. It also shows that the exploitability of the resulting strategy will increase due to the time limit, showcasing the necessity of reducing subgame size.
>
>   Regarding computational complexity in GuanDan: it is indeed a primary concern. In our experiments, we employed two Intel(R) Xeon(R) Gold 6242R CPUs @ 3.10GHz, using the multi-process version of MCCFR. The 30s time limit for GuanDan is greater than that of Texas Holdem, allowing for additional iterations. With these specifications, GS2 is efficient enough for k=10. However, for substantially larger $k$ values, like 40 or 50, convergence becomes problematic in scenarios with expansive branching. Specifically, we noticed minimal improvement against the two baselines when $k=40$ or $50$, compared to the performance when $k=30$. We attribute this to the linear growth in the number of opponent infosets as $k$ increases, causing the strategy to stall before surpassing the performance of the smaller $k$

---

> > ### Comment · Reviewer_gWJJ · 2023-08-12
> >
> > Q1: I think this should be explicitly stated, even perhaps in the body. It greatly changes the interpretation of what the algorithm is doing---assuming the opponent knows your private cards, even temporarily, can greatly change how the subgame solver behaves.
> >
> > Q2: Thank you. I suggest including these in the paper itself.
> >
> > Q3: Thank you. One more follow-up here: your Goofspiel experiments suggest that, at least in the game on the left, GS2-R and GS2-D have about the same performance when k=10. You also say that k=10 is computationally feasible in GuanDan. I'd like to see an ablation running GS2-R in GuanDan, with various k-values, epsecially larger ones---that is, some plot like your global response plots within GuanDan. My purpose here is to understand what exactly in practice is the effect of the diversity function. I understand, however, if this is not possible with the time constraint of the discussion phase.
> >
> >  In any case, my opinion of the paper has not significantly changed, and I keep my score.

---

> > > ### Author Response · Authors · 2023-08-13
> > > **Thanks for your suggestions**
> > >
> > > Thank you for your advise. We will make sure to incorporate the details in the paper.
> > >
> > > Regarding the GS2-D in GuanDan. We are grateful to your understanding regarding the time constraints that prevent from completing the new additional experiments. We will manage to present some preliminary experiment results before the discussion deadline.

---

> > > > ### Author Response · Authors · 2023-08-21
> > > > **Preliminary experiment results for GS2-R**
> > > >
> > > > We conduct new experiments about GS2-R with k=10 in GuanDan against the second champion bot. A total of 516 games were completed, with 129 games executed for each of the four positions. The experiment results are presented as follows:
> > > >
> > > > |       Score       | Position 1 | Position 2 | Position 3 | Position 4 | Average  |
> > > > | :----------: | :--------: | :--------: | :--------: | :--------: | :------: |
> > > > |  Blueprint   |   0.0465   |  -0.0465   |   0.0465   |  -0.0465   |   0.00   |
> > > > | GS2-R $k=10$ |  0.00775   |   -0.093   |   0.0388   |   0.031    | -0.00386 |
> > > > | Improvement  |  -0.03875  |  -0.0465   |  -0.0077   |   0.0775   | -0.00386 |
> > > >
> > > >
> > > >
> > > > Although the experiment is insufficient and the variance might be relatively high, the results provide a preliminary indication that GS2-R with k=10 may not yield significant performance enhancements.
> > > >
> > > > We will provide further results and corresponding plots in the revised paper.

---

> > > > > ### Comment · Reviewer_gWJJ · 2023-08-21
> > > > >
> > > > > Interesting. I think including these experiments will be very nice, as they further illuminate the strength of the diversity metric when compared to the more simple random sample.

---

### Official Review · Reviewer_hSoE · 2023-06-29

**Soundness:** 4 excellent
**Presentation:** 3 good
**Contribution:** 3 good
**Rating:** 6
**Confidence:** 4

**Summary:**

The paper proposes a novel subgame resolving algorithm in Extensive-Form games called GS2, which will only sample a portion of subgames and dramatically reduce computation complexity.

**Strengths:**

- The new subgame resolving algorithm GS2 has a theoretical guarantee, as all previous work did.
- The paper has sufficient experiments and also gets great performance in the Guandan game.

**Weaknesses:**

No code was released. Since subgame refinement is really an engineering topic, if possible, I think releasing the code will greatly benefit the whole community to do improvements in the future. Otherwise it will be really hard to reproduce the work since there are too many details in implementing algorithms.

**Questions:**

Listed in weaknesses.

**Limitations:**

Yes the authors have addressed the limitations adequately.

---

> ### Author Rebuttal · Authors · 2023-08-09
>
> We appreciate your recognition of the merits of our work. In response to your concern, we are in the process of refining our code to ensure its clarity and ease of use. Once finalized, we commit to making it publicly accessible.

---

> > ### Comment · Reviewer_hSoE · 2023-08-10
> > **Re: Rebuttal by Authors**
> >
> > Looking forward to your code!

---

### Official Review · Reviewer_D79X · 2023-07-03

**Soundness:** 2 fair
**Presentation:** 3 good
**Contribution:** 3 good
**Rating:** 6
**Confidence:** 5

**Summary:**

The contributions of the paper are twofold:
- On the theoretical hand, the paper introduces a bound on the total increase in exploitability when refining the strategy in a subgame that consider only some of the possible infosets of the adversary in a stochastic way.
	- This bound is based on considering that the exploitability of the refined strategy may increase (in the worst case) due to updating in the worst possible way the strategy for some of the opponent's infosets ($\delta$ term in the bound). This extra exploitability is then weighted for the probability of not sampling such an adversary infoset ($1-\omega$) term and for the probability of having reached that infoset in game ($\pi$ term)
- This theoretical bound is then used to design of the *Generative Subgame Solving (GS2)* algorithm. GS2 samples some of the nodes $h$ in the current infoset of the refining player. To sample the different $h$ that will belong to the subgame that will be refined, a *diversity-based generation function* is used. This function is an heuristics that selects the histories such that the distribution of counterfactual values at the infosets of the opponent at the root of the GS2 subgame are representative of the distribution that there would have been in the 1-KLSS. This allows to avoid to refine the strategy considering too pessimistic or too optimistic "cuts" of the original subgame, even if there is no formal proof that this reduces exploitability

**Strengths:**

- Interesting and novel technical approach, which allows to interpolate between the unsafe solving from Ganzfried and Sandholm to the safe approach of 1-KLSS
- Formalism used is in line with previous works in the field
- Both medium and larger scale experiments. This allows to verify both the correctness and the scalability of the proposed approach
- clear structure of the paper and clear descriptions of the adopted solutions
- an example of how the technique would be applied on a game is presented, further clarifying the concepts

**Weaknesses:**

- Some of the claims presented by the paper are misleading or poorly justified:
	- The bound presented in Proposition 4.1 bounds the possible increase in exploitability from the blueprint to the refined strategy as the worst case scenario in which the refined strategy is maximally losing in case the opponent decides to switch to play only to that infoset.
	- Lines 235-240 claim that GS2 is more suitable for situations in which $\delta$ is already low, differently to traditional unsafe solving techniques. This means that the blueprint is of low quality, and therefore GS2 cannot make things too much worse in terms of exploitability wrt the original (already bad) blueprint. I don't get why such an argument would not apply as it to any other subgame solving technique
	- Line 290-291: "[GS2] also refines the blueprint by selectively focusing on the most relevant portions of the game tree". The introduced diversity generated function only considers the adversary's infosets as counterfactual values. There is no clear connection to how the sampled histories should be the **most relevant** from a strategy refinement perspective
- The large scale experimental setting presented feels disconnected from the techniques presented by the paper:
	- As indicated in Appendix C.2: *Although standard GS2 need to generate the full second-order knowledge infosets, we only consider the nodes in the first-order knowledge infoset.* I interpreted this as the fact that the set of histories $\{h \in S_{top}: \exists I_2: h \in I_2 \land  \exists h' \in I_1, I_2\}$ is not added to the constructed subgame. My opinion is that this approximation is really important to the point that the resulting algorithm should be clearly distinguished from GS2 as presented in the previous parts of the paper. This because while GS2 is a "partial cut" version of 1-KLSS, not adding all nodes in $I_2$ makes the technique possibly much more unsafe, and more similar to a "partial cut" version of the unsafe abstraction techniques from Ganzfried and Sandholm.
- GS2 is an unsafe solving technique: no cases in which such an unsafeness becomes evident are presented, leaving open the question of when such a technique may fail

**Questions:**

Other then asking the authors to share their view/clarify the weakness I listed in the previous section, I add the following questions to clarify specific points of the paper:
- KLSS's explanation (Line 189-192) is unclear. I suggest just keep formal definition which is fine (and not the partially repeated phase at line 189). Also the $k$-th order knowledge limited subgame starts from the *infoset $I$* of the current player and not from the *current history $h$* as indicated.
- I'd like to ask for a confirmation: is $\pi^{\sigma'_1}_{-2}(\mathcal I_2(h)) = \pi^{\sigma_1}_{-2}(\mathcal I_2(h)) \ \forall h \in S_{top}$?

Minor typos:
- Line 255: is there any connection that was intended between controlling the $k$ parameter and the possible use of abstraction techniques? In my understanding, they are two techniques that can be applied independently. If this is the case, "Therfore" should me substituted with "Moreover" or a synonym thereof.

---

> ### Author Rebuttal · Authors · 2023-08-09
>
> Thank you for your insightful comments. We sincerely appreciate your valuable feedback. Here are our responses.
>
> - **Regarding the exploitability bound.**
>
>   Thank you for pointing out this aspect. Indeed, the bound presented is specific to the infoset. The reason for this is rooted in the fact that GS2 is constructed upon the KLSS framework. Consequently, obtaining an exploitability bound is not as straightforward as with conventional subgame solving techniques. We recognize that the proposition can be extended to strategies where GS2 is applied to disjoint subgames; in such cases, the additional term would vary based on the maximum across these subgames. We will ensure that this clarification is included in the revision of the paper.
>
> - **Regarding L235-240.**
>
>   Thank you for raising this concern. In many unsafe subgame solving techniques, like endgame solving, the assumption is that the opponent blueprint is close to equilibrium; deviations from this can hinder strategy refinement. Thus, the coefficient $1-\omega$ in the exploitation bound will increase up to 1. That's why we say GS2 is more suitable than unsafe solving.
>
> - **Regarding Line 290-291**
>
>   Thanks for the great point. Our use of the term "relevant" was imprecise. What we intended to convey is that the GS2 framework focuses on portions of the game tree that are "typical" or representative based on the adversary's infosets as counterfactual values. We appreciate the clarification and will correct this in the revision to more accurately reflect our approach.
>
> - **Regarding the implementation in Appendix C.**
>
>   We appreciate the observation. The approximation is implemented based on the concept of not just sampling from opponent infosets but also within the infoset itself to manage complexity. This approach is like applying a specific generation function to the opponent's infoset. It leads to a conservative strategy refinement by assuming opponents are aware of the player's exact private information. Such an assumption prevents a drastic rise in exploitability at the present infoset.
>
>   This method is influenced by patterns observed in the game of GuanDan. In this game, as actions unfold, uncertainty diminishes,  thus human players are expected to guess the others' cards especially near the end of the game, which is crucial for expert players. Therefore, considering the worst case that the opponents knows the player's cards would be practical. Even though this idea seems somewhat domain-specific, we believe it is also applicable to similiar games.
>
> - **Regarding the unsafeness.**
>
>   GS2's safety is contingent upon the quality of the generation function used. Specifically, if the generation function were to primarily target situations where opponents face challenges, the refinement could become overly optimistic, making it susceptible to exploitation.
>
> - **Regarding Question 1**.
>
>   Thanks for your suggestions. We will correct it in the later version.
>
> - **Regarding Question 2**.
>
>   If $\sigma_1$ and $\sigma_1'$ are the same outside the subgame $S$, the answer is yes.
>
> - **Regarding the typos.**
>
>   Thanks for the advise. We will make the corrections.

---

> > ### Comment · Reviewer_D79X · 2023-08-18
> >
> > Thanks for your answer, things are clearer now.
> >
> > On top on the minor revisions you already included I strongly suggest to include the discussion on appendix C raised from me and reviewer GWJJ in the main body, as this is crucial to get the whole picture.
> >
> > If I get confirmation of this, I'll raise my score to a 6

---

> > > ### Author Response · Authors · 2023-08-19
> > >
> > > Thank you for your insightful suggestions. We hereby make our commitment to incorporating the comprehensive discussions concerning the implementations in appendix C and other minor revisions discussed before within the main body of revised paper.

---

### Official Review · Reviewer_EqAE · 2023-07-26

**Soundness:** 4 excellent
**Presentation:** 3 good
**Contribution:** 2 fair
**Rating:** 6
**Confidence:** 4

**Summary:**

The paper presents a generative subgame solving framework that can scale to games with a large amount of hidden information. One of the key ideas behind the generative framework is to prioritize exploration based on diversity. The paper evaluates on small-sized tabular games and a large poker-like game called GuanDan.

**Strengths:**

Overall, I have a positive opinion of the paper, with a few reservations (see below). In terms of strengths, I appreciated experiments on a real game. The structure and organization is also appropriate, with only minor clarity issues. While the method appears to be mostly a heuristic, it seems to be working pretty well in practice, and for that reason it seems worthy of discussion at the conference.

**Weaknesses:**

I find that the paper could be improved by expanding the discussion along the following directions:
- I think more should be done to ablate the choice of diversity-based generating function. The method proposed fundamentally boils down to sampling possible compatible histories in the infoset by means of a "diversity-based generation function". Many choices of prioritization could be imagined, and it would be important to have some form of data points about what tends to work and what doesn't. Also, data supporting the need of using prioritization as opposed to not prioritizing as all would be nice to have.
- The discussion around KLSS as introduced by Zhang and Sandholm (L37) seems to suggest that it is a safe method, as it is said "This approach enables the safe refinement of strategies". However, KLSS is typically used in an unsafe way in practice.

I also think that the paper should discuss the following relevant related literature:
- Approaches used in the game of Hanabi. While common-interest in nature, the techniques that have been proposed in Hanabi have an overlap with the material of the paper. For example, the work on Learned Belief Search should be discussed.
- Approaches used in the game of Bridge, such as joint policy search, also seem relevant.

Other comments:
- I feel like calling the games of section 5.1 "medium" is rather generous. The games only have a few hundreds sequences per player and can, for example, be solved exactly using the simplex algorithm via the sequence-form.
- L113, "one cannot assume rationality of the chance player". I found this obscure.

In conclusion, while I find the underlying idea rather straightforward, I think the strengths outweigh the weaknesses despite the limited evaluation. However, I think the paper would substantially benefit from adding more data regarding the choice of prioritization functions, and expanding the discussion regarding related literature.

**Questions:**

Please see above.

**Limitations:**

The authors adequately addressed the limitations.

---

> ### Author Rebuttal · Authors · 2023-08-09
>
> Thank you for your valuable and insightful comments.
>
> - **Regarding the choice of functions.**
>
>   We appreciate your insight on the importance of thoroughly ablating the choice of our diversity-based generation function. In the current study, our choice for the diversity-based generation function was driven by both theoretical considerations and empirical performance in our preliminary experiments. However, we understand the importance of showcasing a more exhaustive comparison. To address this comment, we have included a more detailed comparative analysis in the supplementary material attached to our global response.
>
>   The experiment is conducted in Leduc poker. In this evaluation, we employ all feasible deterministic generations, as opposed to a probability distribution, for each individual infoset right after the cards are dealt. The data points, located at $(x, y, z)$, correspond to instances of generated subgame with the opponent private cards set is $\{x,y,z\}$. The results show that the diversity generation function could perform well at all the player's infoset.
>
> - **Regarding the discussion of KLSS.**
>
>   Thank you for the observation. To clarify, we will revise the statement to read: "This approach can enable safe strategy refinement under specific conditions of usage" in the updated version.
>
> - **Regarding the relevant related literature.**
>
>   Thank you for highlighting the relevance of methods from the games of Hanabi and Bridge. In subsequent versions of our paper, we will expand upon this in the related work section:
>
>   Search algorithms are also applied for obtaining better joint policy witin teammates in collaborative games such as Hanabi [1-2] and the bidding phases of contract Bridge [3].  For example, SPARTA [1] utilizes exact belief update for single agent search and multi-agent search and retrospective belief updates for multi-agent search to handle the large belief range. Subsequently, the strategy is improved through Monte Carlo rollouts. On the other hand, instead of maintaining an explicit belief distribution in single agent search, Learned Belief Search [2] method uses a learned belief model to sample states from the belief distribution, allowing for the application in games with large belief space. Rather than simply improve the strategy via rollouts, Joint Policy Search [3] method first decompose the global changes of the game value into local changes at each infoset, and then iteratively improve the strategy based on the decomposition.Although these approaches primarily focus on improving strategy in collaborative settings and could not be directly used in games with adversaries, the underlying idea could be helpful when developing new technique that conducts search within teammates in games like GuanDan.
>
> - **Regarding the word "medium".**
>
>   Thank you for pointing out the potential mischaracterization. We utilized the term "medium" based on the terminology from KLSS [Zhang and Sandholm]. We will revise the term to "research games" or "simple games" in the updated version of our paper.
>
> - **Regarding the comments on L113.**
>
>   Thank you for pointing out the ambiguity. In L113, when we mention "rationality of a player", we refer to the assumption made in some prior techniques (e.g., the iteratively deepening method used in KLSS) that the opponent will avoid empirically suboptimal actions. This allows one to prune parts of the game tree based on this assumption. However, since a chance player operates under a fixed probability distribution, this assumption of rationality is not applicable.Thus, using the iteratively deepening approach in this context would be unsound. We will elaborate on this distinction in the revised version for clarity.
>
> [1] Lerer A, Hu H, Foerster J, et al. Improving policies via search in cooperative partially observable games[C]//Proceedings of the AAAI Conference on Artificial Intelligence. 2020, 34(05): 7187-7194.
>
> [2] Hu H, Lerer A, Brown N, et al. Learned belief search: Efficiently improving policies in partially observable settings[J]. arXiv preprint arXiv:2106.09086, 2021.
>
> [3] Tian Y, Gong Q, Jiang Y. Joint policy search for multi-agent collaboration with imperfect information[J]. Advances in Neural Information Processing Systems, 2020, 33: 19931-19942.

---

> > ### Comment · Reviewer_EqAE · 2023-08-18
> > **Thanks**
> >
> > Thanks for your response! It all sounds good and I have no further questions at this time.

---

### Author Rebuttal · Authors · 2023-08-09

### **Global response**

Dear Reviewers,

Thank you very much again for your helpful comments. We appreciate your recognization of our work and would like to engage with you in our responses to your questions/comments. If you have any questions about our work or our response, we would be happy to discuss them further.

Best Regards,

the Authors.

---

### Decision · Program_Chairs · 2023-09-21

**Decision:**

Accept (poster)

**Comment:**

The paper proposes a novel sub-game solver that can scale to extensive form games with a large amount of hidden information.
This method proposed in the paper is sound and very effective in practice, as witnessed by its performance in a real game.
The reviewers raised some issues the authors properly addressed in their rebuttals, thus making the reviewers converge on a consensus about acceptance.
The authors need to consider the reviewers' suggestions while preparing the final version of their paper.